# Machine learning for differentiating lung squamous cell cancer from adenocarcinoma using Clinical-Metabolic characteristics and 18F-FDG PET/CT radiomics

Yalin Zhang[1,2], Huiling Liu[3], Cheng Chang[4], Yong Yin[5]*, Ruozheng Wang[1,2]*

1 Department of Radiation Oncology, The Third Affilated Teaching Hospital of Xinjiang Medical University, Affilated Cancer Hospital, Urumuqi, China, 2 Xinjiang Key Laboratory of Oncology, Urumqi, China, 3 Department of Radiation Oncology, Binzhou People's Hospital, Binzhou, China, 4 Department of Nuclear Medicine, The Third Affilated Teaching Hospital of Xinjiang Medical University, Affilated Cancer Hospital, Urumuqi, China, 5 Department of Radiation Oncology, Shandong Cancer Hospital and Institute, Shandong First Medical University and Shandong Academy of Medical Sciences, Jinan, China

* wrz8526@vip.163.com (RW); yinyongsd@126.com (YY)

## Abstract

Noninvasive differentiation between the squamous cell carcinoma (SCC) and adenocarcinoma (ADC) subtypes of non-small cell lung cancer (NSCLC) could benefit patients who are unsuitable for invasive diagnostic procedures. Therefore, this study evaluates the predictive performance of a PET/CT-based radiomics model. It aims to distinguish between the histological subtypes of lung adenocarcinoma and squamous cell carcinoma, employing four different machine learning techniques. A total of 255 Non-Small Cell Lung Cancer (NSCLC) patients were retrospectively analyzed and randomly divided into the training (n = 177) and validation (n = 78) sets, respectively. Radiomics features were extracted, and the Least Absolute Shrinkage and Selection Operator (LASSO) method was employed for feature selection. Subsequently, models were constructed using four distinct machine learning techniques, with the top-performing algorithm determined by evaluating metrics such as accuracy, sensitivity, specificity, and the area under the curve (AUC). The efficacy of the various models was appraised and compared using the DeLong test. A nomogram was developed based on the model with the best predictive efficiency and clinical utility, and it was validated using calibration curves. Results indicated that the logistic regression classifier had better predictive power in the validation cohort of the radiomic model. The combined model (AUC 0.870) exhibited superior predictive power compared to the clinical model (AUC 0.848) and the radiomics model (AUC 0.774). In this study, we discovered that the combined model, refined by the logistic regression classifier, exhibited the most effective performance in classifying the histological subtypes of NSCLC.

**Funding:** This work was supported by the Science and Technology Foundation of Xinjiang Uygur Autonomous Region(2022E02050). It was also supported by the Special Funds Project of Central Guidance on Local Science and Technology Development (ZYYD2022B18).

**Competing interests:** The authors have declared that no competing interests exist.

## Introduction

According to GLOBOCAN 2020, lung cancer ranks as the second most common type of cancer and stands as the leading cause of cancer-related deaths. Approximately 2.2 million new cases were diagnosed in 2020 alone, with the disease accounting for an estimated 1.8 million fatalities [1]. Various types of lung cancer exist, with non-small cell lung cancer (NSCLC) being the most prevalent, constituting about 85% of all lung cancer cases globally [2]. Squamous cell carcinoma (SCC) and adenocarcinoma (ADC) represent the two most common histologic subtypes. Research indicates significant variances in the genetic and epigenetic traits of ADC and SCC during tumorigenesis and progression [3]. Given the differing treatment approaches for adenocarcinoma and squamous cell carcinoma, swift and precise identification of these pathological subtypes is critical.

Approximately one-third of patients diagnosed with NSCLC are at Stage III, a stage at which most are no longer viable candidates for surgical intervention [4]. Consequently, the adoption of computed tomography (CT)-guided biopsies has become the gold standard for determining the pathologic subtype of lung cancer. However, this invasive method may not fully capture the entire tumor's heterogeneity. Given that biopsies typically yield only a few small tissue samples, they may not provide a comprehensive understanding of the overall tumor, posing challenges for accurate diagnosis. Additionally, potential risks associated with biopsy procedures, such as pneumothorax, intrathoracic hemorrhage, pleural reaction, air embolism, and intrapleural implantation metastasis, exist. The prospect of additional biopsies due to heterogeneous or necrotic tumor tissue may deter some patients from undergoing a biopsy, particularly those with an uncontrolled cough [5, 6]. Therefore, the development of a reliable, non-invasive, and practical method for predicting NSCLC histology prior to treatment is paramount.

Relevant studies suggest that certain clinical characteristics can aid in differentiating the diagnosis of lung adenocarcinoma from lung squamous cell carcinoma. These include factors such as age, smoking history, tumor diameter, imaging signs, and microvascular density [7–9]. However, the sole reliance on clinical features for classifying pathological tissues may be influenced by the subjective judgment of the physicians or by the heterogeneity and quantity of the samples, potentially leading to variability in diagnostic outcomes.

As medical and information technologies advance, there is an exponential growth in the volume of medical data, especially in the production of medical imaging. These images harbor extensive latent details pertinent to human health. Yet, the manual examination and interpretation of such data are not only time-consuming but also prone to human bias. Leveraging the capabilities of machine learning can significantly alleviate these issues by extracting sophisticated features and minimizing subjectivity. Radiomics entails the quantitative retrieval of characteristics from conventional medical imaging. The development of predictive or diagnostic models through machine learning techniques allows for the collection of data that can be assimilated into clinical decision-making tools, consequently improving the accuracy of diagnoses or prognosis. [10, 11]. Zhu et al. [12] extracted 485 features from manually delineated tumor regions in 129 NSCLC patients. The results demonstrated that the area under the curve (AUC) for the training and validation sets reached 0.905 and 0.893, respectively, indicating that the imaging features have substantial efficacy in differentiating between lung adenocarcinoma and squamous cell carcinoma. Bashir et al. [13] analyzed the effectiveness of a random forest model utilizing CT image radiomics features, CT semantic features, and combined features in distinguishing lung adenocarcinoma from squamous cell carcinoma. The findings revealed that the random forest model based on radiomics features could non-invasively analyze the histological subtypes of NSCLC with an AUC of 1.

18F-fluorodeoxyglucose (FDG) PET/CT, which combines anatomical and metabolic information, is crucial for identifying primary tumors, staging diseases, and assessing treatment success. Radiomics based on PET/CT shows potential in differentiating ADC from SCC. Yan et al's study [14] developed separate models based on PET and CT, as well as a combined PET-CT model. Among these, the combined model exhibited superior performance in predicting ADC, SCC, and metastasis. Additional studies have also discovered that the inclusion of clinical characteristics, such as gender and smoking history, further enhanced the classification performance, achieving an area under the curve (AUC) of 0.859, which surpassed the performance of radiomics alone [15, 16]. When developing a radiomics prediction model, the selection of an appropriate machine learning algorithm can significantly enhance the model's predictive accuracy and stability. Various classifiers, including Support Vector Machine (SVM), Logistic Regression (LR), and Random Forest (RF), have been utilized to construct models in the studies mentioned earlier. The Light Gradient Boosting Machine (LGBM), a model rooted in gradient boosting decision trees (GBDT), shares principles with the XGBoost algorithm but offers several advantages, such as faster training efficiency, lower memory consumption, higher accuracy, and support for parallel learning. To the best of our knowledge, there has been scant research evaluating the efficacy of the LGBM classifier for radiomics models based on 18F-FDG PET/CT that incorporate clinico-metabolic features for differentiating between ADC and SCC in lung cancer. Therefore, the objective of this research was to create and corroborate a superior machine learning (ML) model utilizing PET/CT data to distinguish between SCC and ADC in stage III NSCLC.

## Materials and methods

### Study design

The Ethics Committee of the Affiliated Cancer Hospital of Shandong First Medical University approved the current retrospective study (No. SDTHEC 2023010008). The requirement for written informed consent was waived due to the study's retrospective nature. The workflow of our study is depicted in Fig 1.

### Patients

Data were accessed for research purposes beginning on March 10, 2023. In this study, we selected a cohort of 255 patients diagnosed with non-small cell lung cancer (NSCLC) between September 2018 and May 2022. The inclusion criteria for this study were as follows: (1) pathologically confirmed non-small cell lung cancer (NSCLC); (2) available PET/CT images obtained before treatment; (3) a diagnosis of stage III disease; (4) a single tumor lesion exceeding 1 cm in diameter. The exclusion criteria included: (1) patients who received anti-tumor treatment prior to the PET/CT scan; (2) individuals with a history of other thoracic malignant tumors or systemic malignancies; (3) patients with pathological confirmation of histological subtypes other than ADC or SCC; (4) patients who underwent surgical intervention after their diagnosis.

In the end, the study enrolled 255 patients, who were then randomly divided into two groups: the training cohort, consisting of 177 individuals, and the internal validation cohort, comprising 78 individuals, following a 7:3 distribution. The clinical characteristics of the patients were systematically documented. Furthermore, the researchers measured various PET metabolic parameters, including metabolic tumor volume (MTV), mean standardized uptake value (SUVmean), maximum standardized uptake value (SUVmax), and minimum standardized uptake value (SUVmin). Additionally, the total lesion glycolysis (TLG) was calculated using the formula TLG = SUVmean × MTV [17].

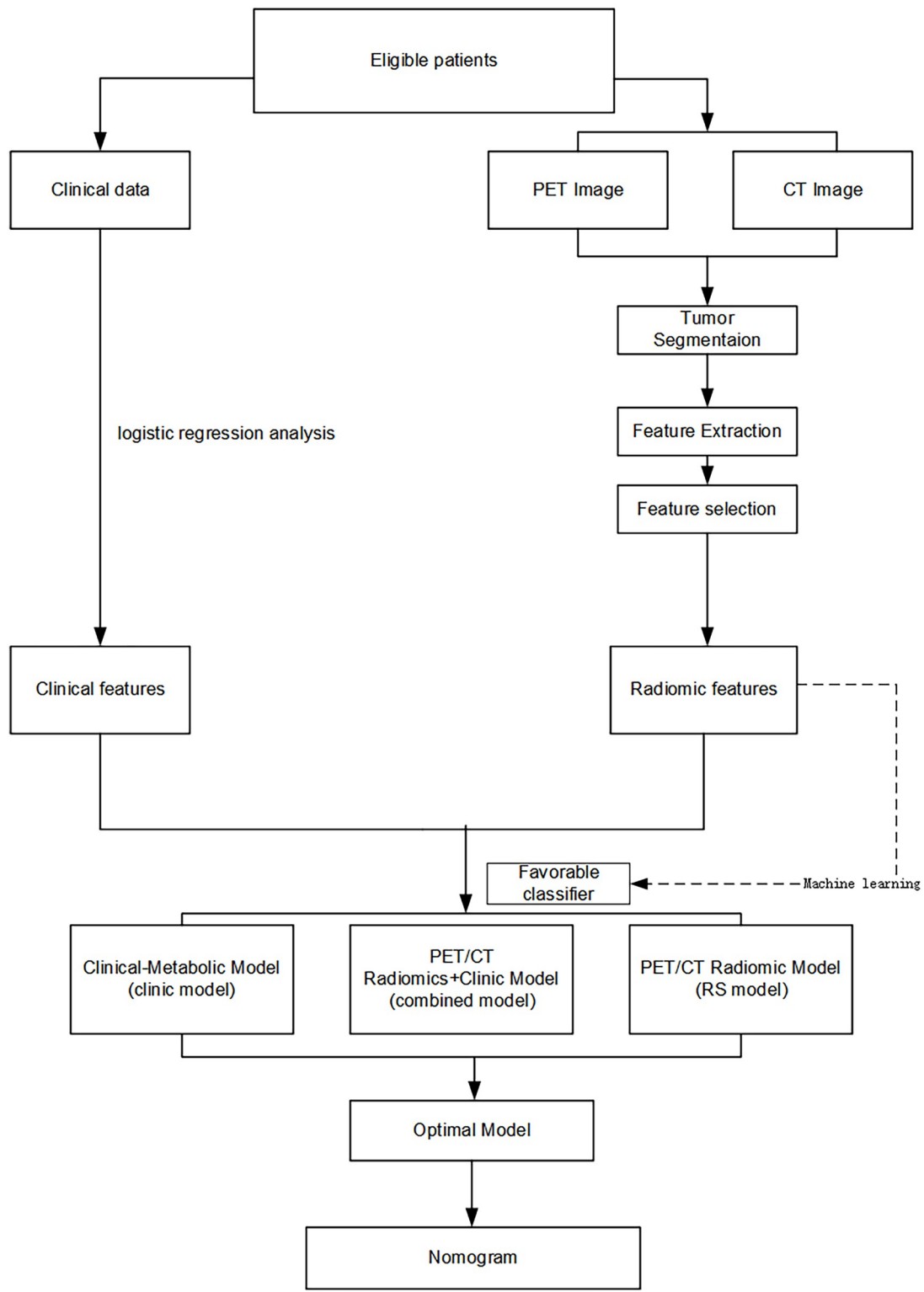

**Fig 1. The workflow of this study.**

## 18F-FDG PET/CT image acquisition

18F-FDG scans were conducted using a Philips Gemini TF PET/CT system (Philips Medical Systems, Netherlands) in accordance with standard clinical scanning protocols. Patients were required to fast for at least six hours prior to the scan, ensuring their blood glucose levels remained below 140 mg/dL. Approximately one hour after the administration of an intravenous dose of 4.4 MBq/kg of 18F-FDG, PET and CT images were acquired. The PET images were reconstructed in multiple planes and reconstruction slice-thickness range of 1 to 3 mm.

## Tumor segmentation

Tumor segmentation was executed using AccuContour software (version 3.2, Manteia Medical Technologies Co., Ltd., Xiamen, China). Two experienced nuclear medicine physicians employed a threshold of 40% of the maximum standardized uptake value (SUVmax) to delineate the gross tumor volume (GTV) on PET images, reaching a consensus without prior knowledge of the pathology [18, 19]. Concurrently, the contours of the GTV on CT slices were outlined based on the integration of PET and anatomical data from the CT images. Subsequently, two senior radiologists conducted a collaborative review of the target images.

## Feature extraction

In this study, the features are divided into three categories: (I) geometric, (II) intensity, and (III) textural. Geometric features capture the three-dimensional shape properties of the tumor. Intensity features reflect the statistical distribution of voxel intensities within the tumor. In contrast, textural features leverage methods such as the gray-level co-occurrence matrix (GLCM), gray-level run length matrix (GLRLM), gray-level size zone matrix (GLSZM), and neighborhood gray-tone difference matrix (NGTDM) to characterize patterns and spatial distributions of voxel intensities. A total of 1,834 handcrafted CT features and 2,016 handcrafted PET features were extracted. All handcrafted features were extracted using a custom feature analysis program implemented in Pyradiomics (http://pyradiomics.readthedocs.io). To integrate PET and CT features, an early fusion approach was employed.

## Feature selection and prediction model establishment

To ensure maximal representation of features while maintaining their distinctiveness, we assessed the correlation among highly repeatable attributes using Spearman's rank correlation coefficient. Features exhibiting a correlation coefficient greater than 0.9 with any other feature were retained. For feature selection, we employed a greedy recursive elimination technique, which systematically removes the most redundant features from the current set at each step. Then, we employed the Least Absolute Shrinkage and Selection Operator (LASSO) regression analysis to select effective features within the training dataset. The LASSO model is particularly effective in reducing regression coefficients toward zero, thereby excluding irrelevant features by setting their coefficients to zero. The initial step involved identifying the optimal regularization parameter, λ. To accomplish this, we utilized 10-fold cross-validation, adhering to the least absolute criterion. By selecting the λ value that minimized the cross-validation error, we identified features with non-zero coefficients that were instrumental in fitting the regression model. These features were then aggregated to construct the radiomic model. Additionally, a radiomics score was computed for each patient by linearly combining the selected features with weights derived from their respective coefficients in the model. The LASSO regression modeling was conducted using the scikit-learn package in Python.

To differentiate between SCC and ADC, three independent predictive models were separately developed: the Clinical-Metabolic Model (clinic model), the PET/CT Radiomic Model (RS model), and the Combined PET/CT Radiomic and Clinical-Metabolic Model (combined model). Four machine learning classifiers, including LR, LGBM, SVM, RF, were used to construct these models. During this process, 5-fold cross-validation was employed to derive these final models.

### Development and validation of individualized nomogram

Furthermore, we constructed a radiomics nomogram using the validation dataset to facilitate a rapid and visual assessment of the enhanced predictive value provided by the combination of the radiomics scores with clinical risk factors. Logistic regression analysis was used in this study to combine radiomic features with clinical risk factors in the nomogram. Finally, we developed calibration curves to appraise the calibration quality of the nomogram.

### Statistical analysis

Patient characteristics were compared using independent sample t-tests, Mann-Whitney U tests, Fisher's exact test, or chi-square ($\chi^2$) tests where relevant. The process of identifying clinical features incorporated both univariate and multivariate logistic regression analyses. The statistical software SPSS (Version 25.0) and R (Version 3.4.0) were used for data analysis, with P-values below 0.05 signaling statistical significance. The selection of the most effective machine learning (ML) model hinged on its performance metrics: the area under the receiver operating characteristic curve (AUC), accuracy (ACC), sensitivity (SEN), and specificity (SPE). AUC comparisons of the various models on the validation set were made using the DeLong test. Decision Curve Analysis (DCA) was also implemented to evaluate the clinical usefulness of the predictive model.

## Results

### Clinical characteristics of patients

This study included a total of 255 participants diagnosed with non-small cell lung cancer (NSCLC). Among these cases, there were 145 patients with squamous cell carcinoma (SCC) and 110 with adenocarcinoma (ADC). The patient population ranged in age from 26 to 85 years, with a mean age of 62 years. The distribution of baseline clinical characteristics was well-balanced between the training and validation cohorts. Table 1 displays the distribution characteristics of the two groups, providing a detailed breakdown of the baseline clinical attributes for each set of patients.

### Features selection and prediction model establishment

To minimize subjective variability in the segmentation of regions of interest (ROI), only radiomic features with both inter-reader and intra-reader Intraclass Correlation Coefficients (ICCs) greater than 0.75 were included. The radiomic features extracted from PET/CT images were categorized into seven distinct groups: first-order features, shape-based features, Gray Level Dependence Matrix (GLDM) features, Gray Level Run Length Matrix (GLRLM) features, Gray Level Size Zone Matrix (GLSZM) features, Neighbouring Gray Tone Difference Matrix (NGTDM) features, and Gray Level Co-occurrence Matrix (GLCM) features. Detailed information about the handcrafted features is provided in Supplementary data (S1 Table). Fig 2 depicts the quantity and distribution of the handcrafted features extracted from the CT and PET images.

**Table 1. Baseline characteristics of patients in cohorts.**

| feature_name | train-label = ALL | train-label = 0 | train-label = 1 | P value | test-label = ALL | test-label = 0 | test-label = 1 | P value |
|---|---|---|---|---|---|---|---|---|
| age | 62.96±8.72 | 61.44±9.55 | 64.19±7.81 | 0.036 | 63.25±8.45 | 61.33±9.18 | 64.47±7.81 | 0.113 |
| smoking | 27.96±29.85 | 21.44±27.11 | 33.29±31.04 | 0.008 | 29.76±31.10 | 28.17±31.91 | 30.78±30.88 | 0.722 |
| BMI | 24.07±3.30 | 23.90±3.17 | 24.21±3.41 | 0.534 | 23.86±3.04 | 23.69±3.28 | 23.96±2.90 | 0.700 |
| WBC | 7.90±2.55 | 7.57±2.70 | 8.17±2.39 | 0.117 | 7.64±2.37 | 7.16±2.01 | 7.94±2.55 | 0.160 |
| NEU | 5.32±2.14 | 5.06±2.26 | 5.53±2.02 | 0.147 | 5.00±1.95 | 4.62±1.63 | 5.24±2.12 | 0.178 |
| LYM | 1.83±0.58 | 1.81±0.63 | 1.84±0.53 | 0.723 | 1.87±0.63 | 1.81±0.68 | 1.91±0.60 | 0.487 |
| EOS | 0.14±0.12 | 0.12±0.11 | 0.15±0.13 | 0.092 | 0.19±0.53 | 0.15±0.14 | 0.22±0.67 | 0.564 |
| BAS | 0.06±0.10 | 0.06±0.10 | 0.05±0.10 | 0.707 | 0.04±0.02 | 0.04±0.02 | 0.04±0.02 | 0.809 |
| PLT | 293.85±89.23 | 294.52±83.49 | 293.31±94.08 | 0.928 | 287.19±93.35 | 272.40±102.35 | 296.64±86.94 | 0.269 |
| CEA | 41.50±175.68 | 84.66±256.10 | 6.28±10.62 | 0.003 | 24.61±74.36 | 43.02±94.11 | 12.85±56.48 | 0.082 |
| LDH | 223.99±85.42 | 228.77±96.36 | 220.08±75.61 | 0.501 | 214.62±57.50 | 216.53±58.25 | 213.40±57.62 | 0.818 |
| ALB | 43.43±10.86 | 43.66±4.48 | 43.24±14.10 | 0.796 | 42.18±4.27 | 42.11±4.47 | 42.23±4.18 | 0.905 |
| Ca2+ | 30.11±12.43 | 29.70±11.78 | 30.45±12.99 | 0.690 | 29.87±12.84 | 27.53±12.26 | 31.36±13.11 | 0.204 |
| NLR | 3.10±1.40 | 3.03±1.45 | 3.15±1.36 | 0.567 | 2.83±1.16 | 2.79±1.26 | 2.86±1.11 | 0.800 |
| TLG | 462.08±605.25 | 317.90±411.69 | 579.78±706.59 | 0.004 | 643.99±747.12 | 452.50±772.18 | 766.22±712.14 | 0.072 |
| SUVmean | 7.41±1.79 | 7.04±1.89 | 7.71±1.65 | 0.013 | 8.21±2.26 | 7.46±1.80 | 8.69±2.41 | 0.019 |
| MTV | 55.39±65.34 | 39.89±46.92 | 68.03±75.09 | 0.004 | 74.61±84.18 | 61.75±94.32 | 82.82±76.96 | 0.287 |
| SUVmax | 14.97±5.85 | 13.27±5.64 | 16.36±5.67 | <0.001 | 17.49±7.51 | 14.16±5.10 | 19.61±8.07 | 0.001 |
| SUVmin | 2.39±0.64 | 2.57±0.58 | 2.25±0.66 | <0.001 | 2.25±0.77 | 2.53±1.07 | 2.07±0.41 | 0.011 |
| dNLR | 2.13±0.82 | 2.11±0.85 | 2.15±0.80 | 0.771 | 1.95±0.66 | 1.93±0.77 | 1.96±0.60 | 0.839 |
| gender | | | | <0.001 | | | | 0.012 |
| 1 | 147(82.58) | 54(67.50) | 93(94.90) | | 67(87.01) | 22(73.33) | 45(95.74) | |
| 2 | 31(17.42) | 26(32.50) | 5(5.10) | | 10(12.99) | 8(26.67) | 2(4.26) | |
| T stage | | | | 0.138 | | | | 0.053 |
| 1 | 19(10.67) | 11(13.75) | 8(8.16) | | 14(18.18) | 9(30.00) | 5(10.64) | |
| 2 | 59(33.15) | 26(32.50) | 33(33.67) | | 21(27.27) | 10(33.33) | 11(23.40) | |
| 3 | 42(23.60) | 23(28.75) | 19(19.39) | | 12(15.58) | 4(13.33) | 8(17.02) | |
| 4 | 58(32.58) | 20(25.00) | 38(38.78) | | 30(38.96) | 7(23.33) | 23(48.94) | |
| N stage | | | | 0.215 | | | | 0.949 |
| 0 | 5(2.81) | 1(1.25) | 4(4.08) | | 2(2.60) | 1(3.33) | 1(2.13) | |
| 1 | 12(6.74) | 3(3.75) | 9(9.18) | | 3(3.90) | 1(3.33) | 2(4.26) | |
| 2 | 87(48.88) | 38(47.50) | 49(50.00) | | 41(53.25) | 15(50.00) | 26(55.32) | |
| 3 | 74(41.57) | 38(47.50) | 36(36.73) | | 31(40.26) | 13(43.33) | 18(38.30) | |
| clinical stage | | | | 0.190 | | | | 0.176 |
| 1 | 56(31.46) | 24(30.00) | 32(32.65) | | 27(35.06) | 12(40.00) | 15(31.91) | |
| 2 | 88(49.44) | 36(45.00) | 52(53.06) | | 34(44.16) | 15(50.00) | 19(40.43) | |
| 3 | 34(19.10) | 20(25.00) | 14(14.29) | | 16(20.78) | 3(10.00) | 13(27.66) | |

WBC, white blood cell count; NEU, neutrophil; LYM, lymphocyte; EOS, eosinophil; BAS, basophil; PLT, platelet; CEA, carcinoembryonic antigen; LDH, lactatedehydrogenase; ALB, albumin; NLR, neutrophil to lymphocyte ratio; dNLR, derived neutrophil to Lymphocyte Ratio; SUVmax, maximum standardized uptake value; SUVmean, mean standardized uptake value; SUVmin, minimum standardized uptake value; TLG, total lesion glycolysis; MTV, metabolic tumor volume; BMI, body mass index; label = 0,ADC; lable = 1 SCC.

Three models were constructed independently using selected clinical factors-metabolic parameters, PET/CT radiomic features, and a combination of the aforementioned variables, utilizing LASSO regression in the training cohort. The clinic model comprised two clinical factors (gender and age), one tumor marker (CEA), and two metabolic parameters (SUVmax and

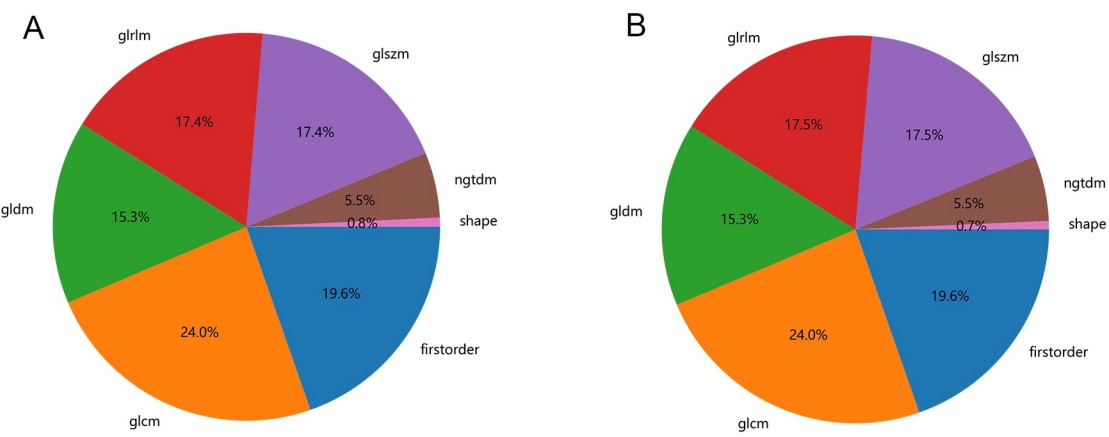

**Fig 2. Number and ratio of handcrafted features.** A show CT features, B show PET features.

SUVmin) (Fig 3A). The analysis suggested that SCC was more prevalent among older males with a long history of smoking, whereas ADC tended to occur in younger females, typically non-smokers (p < 0.05). Univariate logistic regression analysis identified gender, age, smoking history, T stage, white blood cell count (WBC), CEA, total lesion glycolysis (TLG), SUVmin,

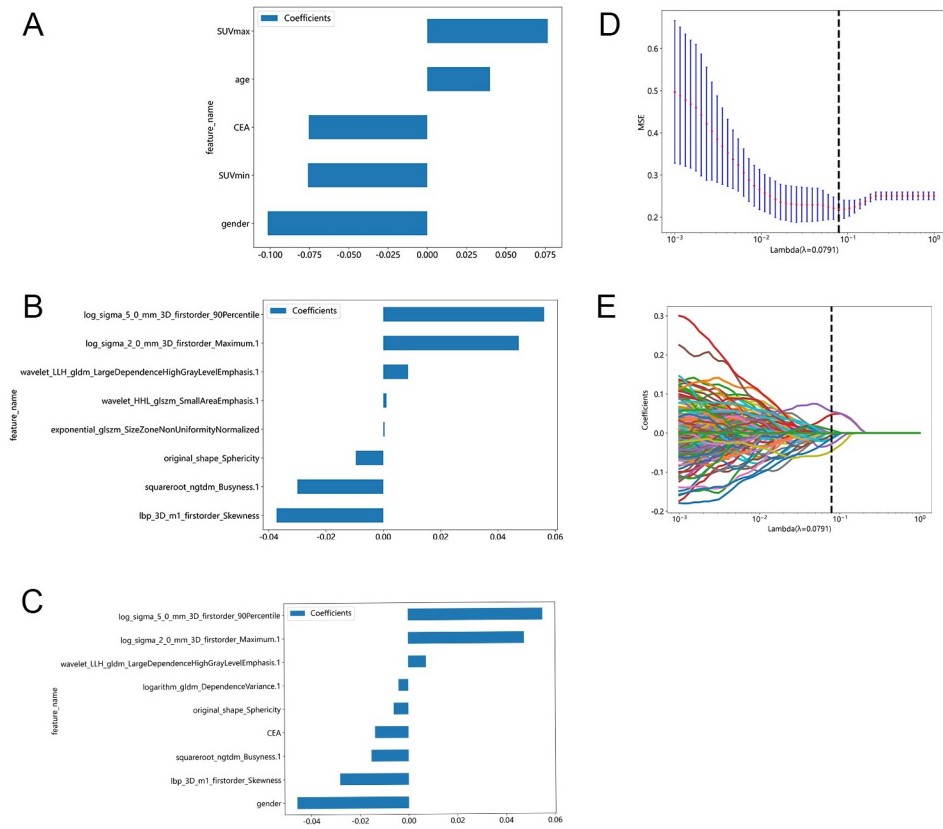

**Fig 3. Radiomic features selected using a LASSO regression model for subgroups.** A-C The coefficients of each feature in the most predictive feature subset. The abscissa is the coefficient, and the ordinate shows the reserved features. The larger the coefficient is, the more predictive effect of the feature is. A shows feature selected in the clinic model, B shows feature selected in the RS model, C shows feature selected in the combined model, D MSE of 10 fold cross validation. E Coefficients of 10 fold cross validation.

SUVmax, SUVmean, and metabolic tumor volume (MTV) as independent risk factors for pathological classification (Table 2, p < 0.05). Multivariate logistic regression analysis confirmed gender, age, CEA, SUVmax, and SUVmin as independent clinical predictors of histology (Table 3, p < 0.05). Within the training cohort for the RS model, eight features were identified (Fig 3B, Table 4). The combined model incorporated two clinical parameters (gender and CEA) along with seven radiomic parameters (Fig 3C–3E, Table 4). To calculate each patient's pre-scores for each model, the following formulas were applied:

$$\text{Pre-score}(\textbf{RS model}) = 0.5393258426966292 + {+}0.000430\times$$
$$\text{exponential\_glszm\_SizeZoneNonUniformityNormalized} - 0.037265\times$$
$$\text{lbp\_3D\_m1\_firstorder\_Skewness} - 0.009572 \times \text{original\_shape\_Sphericity}+$$
$$0.047302 \times \text{log\_sigma\_2\_0\_mm\_3D\_firstorder\_Maximum.1} + 0.056148\times$$
$$\text{log\_sigma\_5\_0\_mm\_3D\_firstorder\_90Percentile} - 0.030010\times$$
$$\text{squareroot\_ngtdm\_Busyness.1} + 0.001106\times$$
$$\text{wavelet\_HHL\_glszm\_SmallAreaEmphasis.1} + 0.008680\times$$
$$\text{wavelet\_LLH\_gldm\_LargeDependenceHighGrayLevelEmphasis.1}$$

$$\text{Pre-score}(\textbf{clinic model}) = 0.5393258426966292 - 0.101583\times$$
$$\text{gender} + 0.039950 \times \text{age} - 0.075441 \times \text{CEA}+$$
$$0.076665 \times \text{SUVmax} - 0.075912 \times \text{SUVmin}$$

$$\text{Pre-score}(\textbf{combined model}) = 0.5393258426966292 - 0.028154\times$$
$$\text{lbp\_3D\_m1\_firstorder\_Skewness} - 0.006010 \times \text{original\_shape\_Sphericity}+$$
$$0.047563 \times \text{log\_sigma\_2\_0\_mm\_3D\_firstorder\_Maximum.1} + 0.055239\times$$
$$\text{log\_sigma\_5\_0\_mm\_3D\_firstorder\_90Percentile} - 0.003944\times$$
$$\text{logarithm\_gldm\_DependenceVariance.1} - 0.015237\times$$
$$\text{squareroot\_ngtdm\_Busyness.1} + 0.007300\times$$
$$\text{wavelet\_LLH\_gldm\_LargeDependenceHighGrayLevelEmphasis.1}-$$
$$0.045824 \times \text{gender} - 0.013716 \times \text{CEA}$$

### Prediction performance and clinical utility of prediction models

Table 5 displays a consolidated overview of the predictive capabilities for differentiating ADC from SCC across multiple machine learning classifiers within the training and internal validation groups. In the validation cohorts, the LR models demonstrated superior outcomes with respect to the AUC, ACC, SEN, and SPE compared to other ML classifiers. As a result, LR was selected as the preferred ML algorithms for the classification of the specified pathological types.

The performance evaluation of the three predictive models using the logistic regression (LR) classifier (Fig 4A and 4B), complemented by the DeLong test results (Table 6), indicated that the combined model surpassed the others, exhibiting superior discrimination and achieving the highest level of accuracy. This was substantiated by the metrics obtained using the

**Table 2. Univariate logistic regression analysis of clinical predictors of histology.**

| feature_name | Log(OR) | lower 95%CI | upper 95%CI | OR | OR lower 95%CI | OR upper 95%CI | p_value |
|---|---|---|---|---|---|---|---|
| gender | -0.474 | -0.605 | -0.343 | 0.622 | 0.546 | 0.710 | 0.000 |
| age | 0.010 | 0.004 | 0.015 | 1.010 | 1.004 | 1.015 | 0.008 |
| BMI | 0.007 | -0.009 | 0.023 | 1.007 | 0.991 | 1.023 | 0.480 |
| smoking | 0.002 | 0.001 | 0.004 | 1.003 | 1.001 | 1.004 | 0.016 |
| T stage | 0.084 | 0.036 | 0.131 | 1.087 | 1.037 | 1.140 | 0.004 |
| N stage | -0.082 | -0.154 | -0.009 | 0.922 | 0.857 | 0.991 | 0.066 |
| clinical stage | -0.010 | -0.082 | 0.063 | 0.990 | 0.921 | 1.065 | 0.827 |
| WBC | 0.025 | 0.005 | 0.046 | 1.026 | 1.005 | 1.047 | 0.042 |
| NEU | 0.028 | 0.003 | 0.052 | 1.028 | 1.003 | 1.053 | 0.061 |
| LYM | 0.038 | -0.049 | 0.125 | 1.039 | 0.952 | 1.133 | 0.471 |
| EOS | 0.119 | -0.047 | 0.285 | 1.126 | 0.954 | 1.330 | 0.239 |
| BAS | -0.180 | -0.785 | 0.425 | 0.835 | 0.456 | 1.530 | 0.623 |
| PLT | 0.000 | -0.000 | 0.001 | 1.000 | 1.000 | 1.001 | 0.607 |
| CEA | -0.001 | -0.001 | -0.000 | 0.999 | 0.999 | 1.000 | 0.001 |
| LDH | -0.000 | -0.001 | 0.000 | 1.000 | 0.999 | 1.000 | 0.448 |
| ALB | -0.001 | -0.006 | 0.005 | 0.999 | 0.994 | 1.005 | 0.782 |
| $Ca^{2+}$ | 0.003 | -0.002 | 0.007 | 1.003 | 0.998 | 1.007 | 0.303 |
| NLR | 0.013 | -0.026 | 0.051 | 1.013 | 0.974 | 1.052 | 0.587 |
| TLG | 0.000 | 0.000 | 0.000 | 1.000 | 1.000 | 1.000 | 0.001 |
| SUVmean | 0.055 | 0.029 | 0.080 | 1.056 | 1.029 | 1.083 | 0.000 |
| MTV | 0.001 | 0.001 | 0.002 | 1.001 | 1.001 | 1.002 | 0.003 |
| SUVmax | 0.023 | 0.015 | 0.030 | 1.023 | 1.015 | 1.030 | 0.000 |
| SUVmin | -0.194 | -0.266 | -0.121 | 0.824 | 0.766 | 0.886 | 0.000 |
| dNLR | 0.010 | -0.056 | 0.076 | 1.010 | 0.946 | 1.079 | 0.799 |

WBC, white blood cell count; NEU, neutrophil; LYM, lymphocyte; EOS, eosinophil; BAS, basophil; PLT, platelet; CEA, carcinoembryonic antigen; LDH, lactatedehydrogenase; ALB, albumin; NLR, neutrophil to lymphocyte ratio; dNLR, derived neutrophil to Lymphocyte Ratio; SUVmax, maximum standardized uptake value; SUVmean, mean standardized uptake value; SUVmin, minimum standardized uptake value; TLG, total lesion glycolysis; MTV, metabolic tumor volume; BMI, body mass index.

**Table 3. Multivariate logistic regression analysis of clinical predictors of histology.**

| feature_name | Log(OR) | lower 95%CI | upper 95%CI | OR | OR lower 95%CI | OR upper 95%CI | p_value |
|---|---|---|---|---|---|---|---|
| gender | -0.388 | -0.526 | -0.249 | 0.679 | 0.591 | 0.780 | 0.000 |
| SUVmin | -0.099 | -0.172 | -0.025 | 0.906 | 0.842 | 0.975 | 0.028 |
| CEA | -0.001 | -0.001 | -0.000 | 0.999 | 0.999 | 1.000 | 0.001 |
| TLG | 0.000 | -0.000 | 0.000 | 1.000 | 1.000 | 1.000 | 0.284 |
| MTV | -0.002 | -0.004 | 0.001 | 0.998 | 0.996 | 1.001 | 0.228 |
| smoking | -0.001 | -0.003 | 0.001 | 0.999 | 0.997 | 1.001 | 0.372 |
| age | 0.008 | 0.003 | 0.014 | 1.008 | 1.003 | 1.014 | 0.009 |
| SUVmax | 0.024 | 0.006 | 0.041 | 1.024 | 1.006 | 1.042 | 0.024 |
| WBC | 0.011 | -0.008 | 0.031 | 1.011 | 0.992 | 1.031 | 0.333 |
| SUVmean | -0.043 | -0.099 | 0.012 | 0.958 | 0.906 | 1.012 | 0.197 |
| T stage | 0.046 | -0.004 | 0.095 | 1.047 | 0.996 | 1.100 | 0.129 |

CEA, carcinoembryonic antigen; SUVmax, maximum standardized uptake value; SUVmean, mean standardized uptake value; SUVmin, minimum standardized uptake value; TLG, total lesion glycolysis; MTV, metabolic tumor volume; WBC, white blood cell count.

**Table 4. The final selected PET-CT radiomics features for RS model and combined model.**

| Model | Filter | Feature class | Feature |
|---|---|---|---|
| RS model | log_sigma_5_0_mm_3D | firstorder | 90Percentile |
| | log_sigma_2_0_mm_3D | firstorder | Maximum.1 |
| | lbp_3D_m1 | firstorder | Skewness |
| | squareroot | ngtdm | Busyness |
| | original | shape | Sphericity |
| | wavelet_LLH | gldm | Large Dependence High Gray Level Emphasis.1 |
| | wavelet_HHL | glszm | Small Area Emphasis.1 |
| | exponential | glszm | Size Zone Non Uniformity Normalized |
| combined model | log_sigma_5_0_mm_3D | firstorder | 90Percentile |
| | log_sigma_2_0_mm_3D | firstorder | Maximum.1 |
| | lbp_3D_m1 | firstorder | Skewness |
| | squareroot | ngtdm | Busyness |
| | wavelet_LLH | gldm | Large Dependence High Gray Level Emphasis.1 |
| | original | shape | Sphericity |
| | logarithm | gldm | Dependence Variance.1 |

RS model, the PET/CT Radiomic Model; combined model, the Combined PET/CT Radiomic and Clinical-Metabolic Model.

**Table 5. Performance of four machine learning algorithms for differentiating pathological subtypes in the training and internal validation cohort.**

| ML | Data set | RS model | | | | |
|---|---|---|---|---|---|---|
| | | AUC | 95% CI | ACC | SEN | SPE |
| LR | Training | 0.806 | 0.743–0.868 | 0.719 | 0.771 | 0.659 |
| | Validation | 0.774 | 0.669–0.879 | 0.727 | 0.694 | 0.786 |
| RF | Training | 1.000 | 1.000–1.000 | 1.000 | 1.000 | 1.000 |
| | Validation | 0.743 | 0.632–0.854 | 0.675 | 0.633 | 0.750 |
| SVM | Training | 0.854 | 0.797–0.911 | 0.787 | 0.792 | 0.780 |
| | Validation | 0.706 | 0.587–0.824 | 0.649 | 0.510 | 0.926 |
| LGBM | Training | 0.880 | 0.832–0.927 | 0.815 | 0.979 | 0.622 |
| | Validation | 0.726 | 0.611–0.842 | 0.688 | 0.592 | 0.857 |

ML, machine learning; RS model, the PET/CT Radiomic Model; CI, confidence interval; ACC, Accuracy; SEN, Sensitivity; SPE, Specificity; LR, logistic regression; SVM, support vector machine; LGBM, light gradient boosting machine; RF, Random Forest.

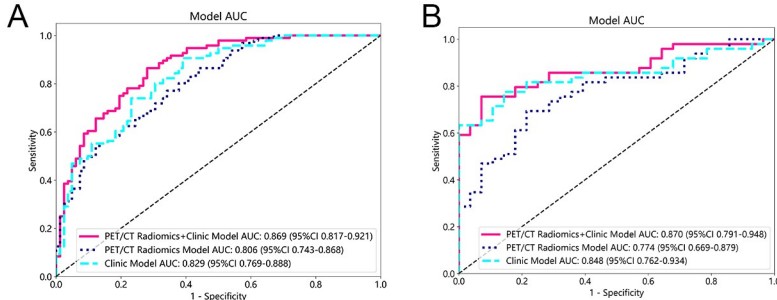

**Fig 4. Comparison of receiver operating characteristic (ROC) curves for predicting subtype of pathology.** A shows the ROC curve of LR in the training cohort; B shows the ROC curve of LR in the validation cohort.

**Table 6. DeLong test within different models based on LR classifier for the validation cohort.**

| LR classifier | | |
|---|---|---|
| **Model 1** | **Model 2** | **P value** |
| combined | RS | 0.042 |
| combined | clinic | 0.013 |
| RS | clinic | 0.001 |

LR, logistic regression; RS, the PET/CT Radiomic Model; combined, the Combined PET/CT Radiomic and Clinical-Metabolic Model; clinic, the Clinical-Metabolic Model.

logistic regression classifier in both the training cohort (AUC (95% CI) = 0.869 (0.817–0.921)) and the validation cohort (AUC (95% CI) = 0.870 (0.791–0.948)), with both p-values being less than 0.05.

A nomogram was developed to integrate clinical and radiomic signatures, and it exhibited the most robust performance (Fig 5A). Decision curve analysis (DCA) further confirmed that the combined model serves as the most dependable clinical instrument for predicting

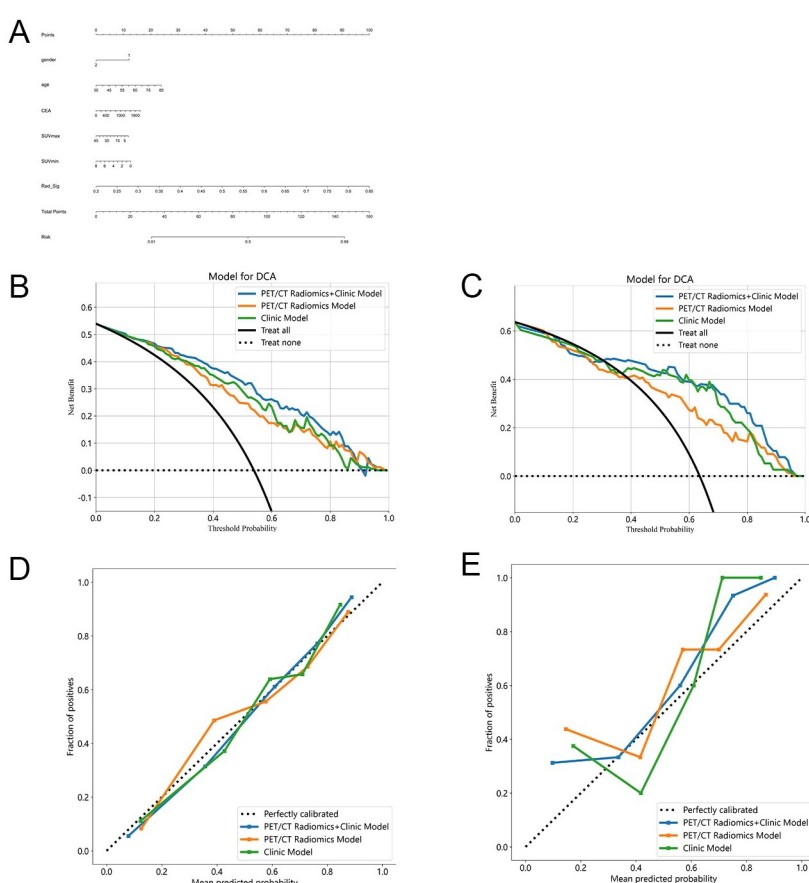

**Fig 5. Clinical utility of prediction models.** A shows Nomogram of a clinical radiomics model developed based on a logistic regression model for the training cohort. gender 1:male 2:female. B,C show that Decision curve analysis (DCA) was conducted for the prediction model based on the logistic regression model in the training (B) and validation cohorts (C). D,E show Calibration curves of the nomogram based on the logistic regression model in the training (D) and validation cohorts (E).

histological subtypes, especially when the threshold probability surpasses 20% (Fig 5B and 5C). Calibration curves for both the training and validation cohorts of the nomogram demonstrated a high degree of agreement between the predicted histology and the actual observations, with the combined model using the logistic regression algorithm showing particular effectiveness (Fig 5D and 5E).

## Discussion

Personalized treatment plays a crucial role in improving patient survival outcomes. At the heart of personalized medicine lies the early and accurate diagnosis and staging of lung cancer, as well as the precise identification of its pathological subtypes. Although biopsy is considered the gold standard for diagnosing lung cancer, its invasiveness, limited reproducibility, possibility of yielding false-negative results, and the associated risk of complications highlight the urgent need for enhanced diagnostic methods. Therefore, the differentiation of pathological subtypes of non-small cell lung cancer (NSCLC) through standard imaging modalities remains a substantial challenge. In this study, we compared four classifier models to identify the pathological subtypes of non-small cell lung cancer. The optimal classifier was evaluated for its predictive efficacy across three models: the RS model, the clinic model, and the combined model. In this study, we discovered that the combined model, refined by the logistic regression classifier, exhibited the most effective performance in classifying the histological subtypes of NSCLC.

We explored the clinical features that contribute to the differentiation between ADC and SCC in NSCLC. We found that gender, age, CEA levels, maximum standardized uptake value and minimum standardized uptake value were statistically significant discriminators between ADC and SCC, which were accordence with other studies. Previous research has validated gender and age as clinical characteristics that can distinguish between ADC and SCC. Koh et al. [20] compared intratumoral stromal proportions and positron emission tomography (PET) textural features in females and males diagnosed with either adenocarcinoma or squamous cell carcinoma. Their findings indicated a higher prevalence of ADC in females compared to males. Additionally, the variation in tumor heterogeneity between women w ith ADC and men with ADC or SCC suggests that gender may serve as a distinguishing feature. Younger patients are more commonly diagnosed with adenocarcinoma, aligning with the findings of several studies [21–24]. This trend may be attributable to the different mutation rates of genes such as EGFR, ALK, and KRAS in younger versus older lung cancer patients. In our cohort, the ADC group consisted of younger individuals than the SCC group (P<0.05), although the average age in both groups exceeded 60 years. This contrasts with other studies that categorize patients as younger if under 40 and older if over 60. The discrepancy can primarily be ascribed to the specific sample population in our research. Serum CEA levels are commonly measured to identify lung cancer, serving as a tumor marker. Elevated CEA levels are seen in 35% to 70% of NSCLC patients, particularly in those with lung adenocarcinoma and advanced disease [25], a finding that our study corroborates. Furthermore, research by Karam et al. [26–28] on 98 NSCLC cases established a significant correlation between SUVmax and the size of primary lesions, with SCC showing notably higher SUVmax values than ADC. Our study confirms that SUVmax is indeed higher in SCC compared to ADC (P < 0.05). We also observed that SUVmin is higher in ADC than in SCC (P < 0.05), adding another layer to the diagnostic criteria for these subtypes.

In addition to analyzing clinical features, this study also integrated radiomic features from PET/CT images. In our study, the most significant radiomic features for both the RS model and the combined model were lbp_3D_m1_firstorder_Skewness,

log_sigma_5_0_mm_3D_firstorder_90Percentile, squareroot_ngtdm_Busyness and log_sigma_2_0_mm_3D_firstorder_Maximum. A previous study [29] demonstrated that first-order features were particularly stable and robust in rectal cancer. In our research, three first-order features were also confirmed to be the most significant indicators for classifying histological types. Another noteworthy feature in our study was 'busyness,' which is associated with the spatial frequency of intensity changes. Erol M et al. [30] reported that radiomic features, including busyness, were independently correlated with the staging of lung squamous cancer. Bashir et al. [13] and Hyun et al. [16] have previously explored the application of radiomics in the classification of NSCLC. However, in their studies, the radiomic features exhibiting the highest performance include separately GLSZMSZLIE, coefficient of variation, NGTDM coarseness and gray-level zone length nonuniformity, gray-level nonuniformity for zone. The best-performing subset of radiological features in our study differs from those identified in other studies [31, 32]. This discrepancy may be attributed to the fact that there are hundreds of radiomic features, many of which are inter-correlated, leading to the possibility that different high-ranking features might essentially represent variations of the same underlying feature.

Regarding the predictive capabilities of different models, several studies [33, 34] have indicated that a combined model incorporating both PET and CT features yields a higher Area Under the Curve than models using only PET or CT features individually. Ren [35] analyzed preoperative clinical features, tumor markers, and PET and CT imaging characteristics, subsequently constructing four independent predictive models. The DeLong test revealed that the combined model exhibited superior performance in predicting the pathological subtypes of NSCLC, with an AUC of 0.932 for the training cohort and an AUC of 0.901 for the validation cohort. These findings align with those of our study, which determined that the combined model achieved higher AUC values compared to the pure PET-CT model and the clinical model alone.

In the construction of radiomics-based prediction models, the selection of suitable machine learning algorithms can enhance the predictive accuracy and stability of the model. Recent advancements in machine learning algorithms, including Gaussian processes, decision trees, RF, SVM, LGBM, and LR, have propelled the application and development of radiomics. Shen [36] evaluated seven different classifiers to optimize a model for classification: SVM with a linear kernel, SVM with a radial basis function kernel (SVM-RBF), RF, LR, Gaussian process classifier (GP), linear discriminant analysis (LDA), and the AdaBoost classifier. The study found that a PET/CT radiomics model using the SVM-RBF classifier demonstrated the best performance, with an AUC of 0.9155, when integrating subregion imaging from PET-CT scans and clinical features for classifying histological subtypes of NSCLC. Parmar et al. [37] demonstrated that the random forest method was the most effective in managing radiomic feature instability, outperforming 12 other machine learning classifiers, including bagging, Bayesian, boosting, decision trees, discriminant analysis, generalized linear models, multiple adaptive regression splines, nearest neighbors, neural networks, partial least squares and principal component regression, and SVMs—in terms of prognostic performance. Huang, et al. [38] employed the LGBM algorithm to develop both a radiomic model and a fusion model (clinical + radiomic) to predict EGFR mutation status in patients with NSCLC. Models based on radiomic signatures can provide relatively accurate non-invasive predictions of EGFR expression status.

In our study, we found that the combined model constructed using the Logistic Regression algorithm performs excellently in identifying pathological subtypes, even when compared with multiple algorithms, including Light Gradient Boosting Machine. Moreover, LGBM also demonstrated good predictive performance. Logistic regression, a linear model, is widely favored for binary classification problems due to its computational simplicity and interpretability. LR

is adept at handling large datasets and excels with linearly separable problems. Our finding aligns with a similar study conducted by Ren et al. [35], which reported that a combined model incorporating clinico-biological features and 18F-FDG PET/CT data, utilizing the LR algorithm, showed strong capability in distinguishing SCC from ADC. Additionally, another study suggested that a model trained on 18F-FDG PET radiomics with the LR algorithm could be effective for predicting the histological subtypes of lung cancer [16]. Given that different machine learning algorithms have their respective optimal application contexts, it is imperative to explore a variety of algorithms to identify the most suitable model for predicting NSCLC histology subtypes. For instance, Gao et al. proposed an improved adaptive neuro-fuzzy inference system-based machine learning method to predict the multi-axis fatigue life of various metal materials. The study found that this model exhibited superior predictive performance and extrapolation capabilities when compared with six classical machine learning models [39]. Moreover, a recent study indicated that a 3D convolutional neural network (CNN) model effectively differentiated between benign and malignant pulmonary nodules in 2-[18F] FDG PET images [40]. The optimization of machine learning algorithms should be prioritized in future research to improve the performance of predictions.

Our study has several limitations. Firstly, this study is based on data from a single center and only includes a training set and a inner validation set. Multi-center data can be included to enhance the stability of predictions. Secondly, the classification method adopted in this study is machine learning. In the future, we can attempt to incorporate deep learning techniques, even with multiomics to optimize the classification model.

## Conclusion

In this study, we developed a comprehensive model that integrates clinical characteristics with PET/CT imaging features using the logistic regression algorithm. This model serves as an effective tool for the "virtual biopsy" of stage III non-small cell lung cancer, distinguishing between different pathological subgroups. It can aid physicians in making informed clinical decisions concerning treatment options and prognostic assessments.

## Supporting information

**S1 Table. Radiomics feature extraction.**
(DOC)

**S1 Dataset. PET-CT radiomics features.**
(ZIP)

**S2 Dataset. Baseline characteristics of patients.**
(ZIP)

**S1 File.**
(DOCX)

**S2 File.**
(PDF)

## Acknowledgments

Thanks to our colleagues at the Department of Radiation Oncology, Shandong Cancer Hospital. Thanks for the technical support provided by Onekey AI platform.

## Author Contributions

**Conceptualization:** Yalin Zhang, Yong Yin, Ruozheng Wang.

**Data curation:** Cheng Chang.

**Formal analysis:** Yalin Zhang, Huiling Liu.

**Funding acquisition:** Cheng Chang, Ruozheng Wang.

**Methodology:** Yalin Zhang, Huiling Liu.

**Project administration:** Yong Yin, Ruozheng Wang.

**Software:** Huiling Liu.

**Supervision:** Ruozheng Wang.

**Validation:** Huiling Liu.

**Writing – original draft:** Yalin Zhang.

**Writing – review & editing:** Yong Yin, Ruozheng Wang.

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
