## [Decision Letter · Decision Letter 0]

21 Dec 2023

PONE-D-23-34795Machine Learning for Differentiating Lung Squamous Cell Cancer From Adenocarcinoma Using Clinical-metabolic Characteristics and 18F-FDG PET/CT RadiomicsPLOS ONE

Dear Dr. Wang,

Thank you for submitting your manuscript to PLOS ONE. After careful consideration, we feel that it has merit but does not fully meet PLOS ONE’s publication criteria as it currently stands. Therefore, we invite you to submit a revised version of the manuscript that addresses the points raised during the review process.

We look forward to receiving your revised manuscript.

Kind regards,

Francesco Dondi

Academic Editor

PLOS ONE

Journal Requirements:

Reviewers' comments:

Reviewer's Responses to Questions

**Comments to the Author**

1. Is the manuscript technically sound, and do the data support the conclusions?

Reviewer #1: Yes

Reviewer #2: No

Reviewer #3: Yes

2. Has the statistical analysis been performed appropriately and rigorously? 

Reviewer #1: Yes

Reviewer #2: N/A

Reviewer #3: Yes

3. Have the authors made all data underlying the findings in their manuscript fully available?

Reviewer #1: Yes

Reviewer #2: Yes

Reviewer #3: Yes

4. Is the manuscript presented in an intelligible fashion and written in standard English?

Reviewer #1: No

Reviewer #2: No

Reviewer #3: Yes

5. Review Comments to the Author

Reviewer #1: Following are the suggestions and questions:

1.The abstract is detailed enough but not concise enough. What's more, it should not contain subtitles.

2.Relevant research is not sufficiently introduced. Please provide a detailed introduction to the progress in building predictive models based on radiomics and clinical data. Relevant research progress on other diseases can also be introduced.

3.At the same time, the highlights and innovations of this article also need to be elaborated on in the Introduction.

4.The structure of this paper is confusing. Especially in the main text since Materials and Methods, there are no title numbers and subtitles, making it difficult for readers to read.

5.Are the Rad signature and Clinical signature mean the prediction values based on radiomic features and clinical features respectively? Does Rad refer to Radiomic? If so, please explain in the article or use Radiomic instead of Rad.

6.The selection of features is crucial to the machine learning algorithm. It is recommended that the selected handcrafted features extracted from CT and PET images be attached and elaborated in detail.

Reviewer #2: Althouth , my comments mights hurt the authors , I appretiate your intrest in this chosen feild.

However, article is not in in a threshold to recommend for further consideration, therefore I forced to make my remark to Reject this manuscript.

Like in the manuscript abstract section, starting from purpose then conclusion which is not clear , you have to make it in a clearly understandable way.

Also you have mentioned about "1,834 handcrafted CT features and 2,016 handcrafted PET features" , since your model is AI based what is the rational to choose this? Hoew many from each data? How can you make it in a stantard format from each data? "Lung cancer has a variety of types.Example from introduction: see this sentense : " . This is kind of incomplete sentense, you have to correct almost every sentenses.

Anyway, these mentionsd are not the solid reasons for my remark , throughout the draft need a rebuild. Figures and tables are not in conicse form. Overall , I recommend authors to make much attention and serious prepration.

Reviewer #3: In this manuscript, The clinical, radiomics, and combination models were developed using two different types of machine learning classifiers. The machine learning approaches were proposed for differentiating lung squamous cell cancer from adenocarcinoma using clinical-metabolic characteristics and 18F-FDG PET/CT radiomics. This manuscript is well-organized with associated results clearly presented. Specific comments are as follows:

(1)The abstract of this manuscript should be refined.

(2) Why the authors adopt machine learning methods for research should be clearly explained in the manuscript.

(3) A technical roadmap for this study is suggested to be added in this manuscript.

(4) The principle of the machine learning method used in this manuscript should be introduced in detail.

(5) The comparison analysis between the proposed method and the existing methods is expected in this manuscript.

(6) More recent works about machine learning should be included in this manuscript, like “A novel machine learning method for multiaxial fatigue life prediction: Improved adaptive neuro-fuzzy inference system. International Journal of Fatigue, 2024, 178: 108007.”

(7) In this manuscript, the features selection and prediction model establishment should be described in a more detailed approach.

6. PLOS authors have the option to publish the peer review history of their article (what does this mean?). If published, this will include your full peer review and any attached files.

Reviewer #1: No

Reviewer #2: No

Reviewer #3: No

---

## [Author Response · Author response to Decision Letter 0]

4 Feb 2024

We would like to thank the reviewers and editor for their insightful and constructive feedback. We feel that the referees’ input has enabled us to produce a greatly improved manuscript, whose novelty and impact are now more readily apparent in the introduction and discussion. Based on the reviewers’ comments, we have made extensive alterations to the structure, format, presentation, and analysis of our findings, with particular attention to adhering to the standards of academic English.

The reviewer specifically raised some concerns regarding the novelty and significance of our work. In response,we have reformatted the introduction to more clearlyaddress a broader swath of the novelty of our work within the appropriate context in both the introduction and discussion sections. In addition, the reviewers also expressed concern about several technical issues. In response, we have addressed all of the reviewer’s technical concerns with a substantial amount of new data, supported by additional detailed explanations. The comments of the reviewers are response point by point and the revisions are indicated in attach files.

---

## [Decision Letter · Decision Letter 1]

23 Feb 2024

Machine Learning for Differentiating Lung Squamous Cell Cancer From Adenocarcinoma Using Clinical-metabolic Characteristics and 18F-FDG PET/CT Radiomics

PONE-D-23-34795R1

Dear Dr. Wang,

We’re pleased to inform you that your manuscript has been judged scientifically suitable for publication and will be formally accepted for publication once it meets all outstanding technical requirements.

Kind regards,

Francesco Dondi

Academic Editor

PLOS ONE

Additional Editor Comments (optional):

Reviewers' comments:

Reviewer's Responses to Questions

**Comments to the Author**

1. If the authors have adequately addressed your comments raised in a previous round of review and you feel that this manuscript is now acceptable for publication, you may indicate that here to bypass the “Comments to the Author” section, enter your conflict of interest statement in the “Confidential to Editor” section, and submit your "Accept" recommendation.

Reviewer #1: All comments have been addressed

Reviewer #3: All comments have been addressed

2. Is the manuscript technically sound, and do the data support the conclusions?

Reviewer #1: Yes

Reviewer #3: Yes

3. Has the statistical analysis been performed appropriately and rigorously? 

Reviewer #1: Yes

Reviewer #3: Yes

4. Have the authors made all data underlying the findings in their manuscript fully available?

Reviewer #1: (No Response)

Reviewer #3: Yes

5. Is the manuscript presented in an intelligible fashion and written in standard English?

Reviewer #1: Yes

Reviewer #3: Yes

6. Review Comments to the Author

Reviewer #1: In the Patients section, how are the training set and the test set randomly divided? Is cross-validation used?

Reviewer #3: I think the paper is ready for publication as it is. Authors have adequately addressed all my comments.

7. PLOS authors have the option to publish the peer review history of their article (what does this mean?). If published, this will include your full peer review and any attached files.

Reviewer #1: No

Reviewer #3: No

---

## [Editor Report · Acceptance letter]

21 Mar 2024

PONE-D-23-34795R1 

PLOS ONE

Dear Dr. Wang, 

I'm pleased to inform you that your manuscript has been deemed suitable for publication in PLOS ONE. Congratulations! Your manuscript is now being handed over to our production team.

Kind regards, 

on behalf of

Dr Francesco Dondi 

Academic Editor

PLOS ONE